# Cultural Moderation of Unconscious Hedonic Responses to Food

**DOI:** 10.3390/nu11112832

**Published:** 2019-11-19

**Authors:** Wataru Sato, Krystyna Rymarczyk, Kazusa Minemoto, Jakub Wojciechowski, Sylwia Hyniewska

**Affiliations:** 1Kokoro Research Center, Kyoto University, 46 Shimoadachi, Sakyo, Kyoto 606-8501, Japan; minemoto.kazusa.6w@kyoto-u.ac.jp (K.M.); sylwia.hyniewska@gmail.com (S.H.); 2Department of Experimental Psychology, Institute of Cognitive and Behavioural Neuroscience, SWPS University of Social Sciences and Humanities, 03-815 Warsaw, Poland; 3Bioimaging Research Center, Institute of Physiology and Pathology of Hearing, 02-042 Warsaw, Poland; wiercirurki@gmail.com

**Keywords:** cross-cultural experiment, food, subliminal affective priming, Japanese food, Poland, unconscious emotional response

## Abstract

Previous psychological studies have shown that images of food elicit hedonic responses, either consciously or unconsciously, and that participants’ cultural experiences moderate conscious hedonic ratings of food. However, whether cultural factors moderate unconscious hedonic responses to food remains unknown. We investigated this issue in Polish and Japanese participants using the subliminal affective priming paradigm. Images of international fast food and domestic Japanese food were presented subliminally as prime stimuli. Participants rated their preferences for the subsequently presented target ideographs. Participants also rated their preferences for supraliminally presented food images. In the subliminal rating task, Polish participants showed higher preference ratings for fast food primes than for Japanese food primes, whereas Japanese participants showed comparable preference ratings across these two conditions. In the supraliminal rating task, both Polish and Japanese participants reported comparable preferences for fast and Japanese food stimuli. These results suggest that cultural experiences moderate unconscious hedonic responses to food, which may not be detected based on explicit ratings.

## 1. Introduction

Hedonic or emotional responses to food play vital roles in human well-being (e.g., facilitating life satisfaction and happiness [1]) and ill-being (e.g., triggering overeating and lifestyle-related disease [2,3]). Previous psychological studies have shown that the observation and consumption of food trigger hedonic responses, which in turn motivate the consumption of food [4,5,6]. A recent study further demonstrated that hedonic responses to food occur rapidly, even before the conscious perception of food [7]. The study utilized a subliminal affective priming paradigm [8] and revealed that subliminal presentation of food images facilitated preferences for subsequent target stimuli more than subliminal presentation of scrambled mosaic images did. These findings suggest that hedonic responses to food are triggered both consciously and unconsciously.

A number of cross-cultural psychological studies have shown that people’s cultural experiences moderate conscious hedonic responses while viewing and eating food [9,10,11,12,13,14,15,16,17,18,19,20,21]. Although extant studies rarely define “culture”, cultural differences were generally assumed that cultural differences are related to regional differences and that cultures facilitate the development of food preferences through exposure or familiarity [9,13,16,17,18,19,20,21] as well as through social influences and norms [13]. For example, Prescott et al. conducted a series of cross-cultural food consumption studies in Australian and Japanese participants using several domestic and international food products [22]. Their results generally showed that participants reported higher hedonic ratings for their domestic food (e.g., Japanese food for Japanese participants) than for non-domestic food, and the groups showed comparable ratings for international food. Torrico et al. [21] showed images of various food products to participants with Western and Asian backgrounds and asked them to self-report their preference ratings. The results showed that Western and Asian groups showed higher preference ratings for Western- and Asian-origin food products, respectively. These data suggest that cultural factors moderate hedonic responses to food, heightening preferences for food from one’s own culture.

However, whether cultural experiences moderate unconscious hedonic responses to food remains unknown. This issue is important because some researchers have proposed that the extent to which individuals engage in certain daily eating behaviors is controlled unconsciously [23]. Consistent with this, a previous study showed that unconscious, rather than conscious, preferences for food are associated with daily eating behaviors [7]. Although no research tested this, some previous studies reported that participants’ cultural dispositions moderated the processing of subliminally presented non-food stimuli, such as emotional facial expressions [24,25]. Based on these findings, we hypothesized that participants’ cultural experiences could moderate unconscious hedonic responses to food.

To test this hypothesis, we carried out a study with participants from two cultural backgrounds, Polish and Japanese, using photographs of international fast food (i.e., a hamburger) and domestic Japanese food (i.e., sushi) (Figure 1). To investigate unconscious hedonic responses, we used the subliminal affective priming paradigm used in a previous study [7]. We subliminally presented food images and their scrambled mosaics as prime stimuli, followed by nonsense ideographs as target stimuli; participants rated their preferences for the target stimuli. A number of previous studies using this paradigm found that participants’ evaluations of the target were biased toward a greater preference for positive primes over neutral primes [8,26,27]. This represents evidence of the elicitation of unconscious emotion, which then spills over into the evaluation of unrelated targets [8]. To investigate conscious hedonic responses, we also presented food and mosaic images supraliminally, and the participants rated their preferences for these images. We predicted that Polish participants, but not Japanese participants, would show higher preference ratings in response to the subliminally-presented fast food than to Japanese food primes and to supraliminally-presented fast food than to Japanese food images.

## 2. Materials and Methods

### 2.1. Participants

We tested 29 healthy Polish volunteers (20 females and 9 males; mean ± SD age, 24.1 ± 3.2 years) and 29 healthy Japanese volunteers (19 females and 10 males; mean ± SD age, 23.2 ± 5.0 years). The required sample size for a repeated-measures analysis of variance (ANOVA) with one between- and one within-subjects factor (two levels each) was determined through an *a priori* power analysis using G*Power software ver. 3.1.9.2 [28], assuming an α level of 0.05, a power (1 - *β*) of 0.80, and a repeated-measures correlation of 0.2 (estimated based on our previous data [7]). Because the effect size was unclear, we predicted medium-sized effects (*f* = 0.25). The result of the power analysis showed that more than 54 participants were needed. The participants were recruited by means of advertisements at the SWPS University of Social Sciences and Humanities and Kyoto University, respectively. All Polish and Japanese participants lived in Poland and Japan, and spoke Polish and Japanese, respectively. Furthermore, only Polish participants who do not regularly (more than twice per week) eat Japanese food were tested. Polish and Japanese participants were matched for gender (*X*^2^-test, *p* > 0.10) and age (*t*-test, *p* > 0.10). None of the participants were obese (mean ± SD body mass index (BMI): Polish: 21.8 ± 2.9; Japanese: 21.5 ± 3.0; *t*-test, *p* > 0.10), and none could read Korean characters (the target ideographs). All of the participants had normal or corrected-to-normal visual acuity, and all were blind to the research purpose and had fasted for more than 3 h prior to the experiments. Their hunger levels were assessed before the experiments using a five-point scale ranging from 1 (hungry) to 5 (satiated); the results indicated that they were relatively hungry (mean ± SD: Polish: 2.0 ± 0.6; Japanese: 2.2 ± 0.6; *t*-test, *p* > 0.10). Although an additional three Polish and four Japanese volunteers participated, their data were not analyzed because they reported having consciously perceived food images during the subliminal rating task. After the experimental procedures had been explained, written informed consent was provided by all participants. This study was approved by the Ethics Committee of the Unit for Advanced Studies of the Human Mind, Kyoto University.

### 2.2. Stimuli

Food stimuli were color photographs of fast food (three images for each of four sub-types: hamburgers, fried chicken, pizzas, and doughnuts) and Japanese food (three images for each of four sub-types: sushi, roast fish, Japanese mixed rice, and udon noodles) (Figure 1). The food images were gathered from websites and then cropped and adjusted using Photoshop CS6 (Adobe, San Jose, CA, USA). The size of all food images was 5.0° × 5.0°. The scrambled mosaic stimuli were generated from the food images using MATLAB 6.5 (MathWorks, Natick, MA, USA). For this process, all food images were divided into small squares (40 × 40) and randomly reordered. This rearrangement made each food image unrecognizable. As a mask stimulus, a scrambled mosaic image was created in the same way using a food image not employed in the experiments.

The target ideographic stimuli were 48 Korean characters. We selected these target stimuli, because unfamiliar ideographs have been used in a number of previous subliminal affective priming experiments as ambiguous, emotionally neutral stimuli that can clearly reflect the effect of emotional primes [8,26,27]; several previous studies have demonstrated that these stimuli are emotionally neutral and can reveal the subliminal priming effect [29,30]. The size of all target stimuli was 5.0° × 5.0°.

### 2.3. Apparatus

The experiments were run using Presentation software (Neurobehavioral Systems, Berkeley, CA, USA) on Windows computers (HP Z200 SFF; Hewlett-Packard Japan, Tokyo, Japan). The images were presented on a 19-inch cathode ray tube monitor (HM903D-A; Iiyama, Tokyo, Japan) with a resolution of 1024 × 768 pixels and refresh rate of 100 Hz. The responses were obtained using a response box (RB-530; Cedrus, San Pedro, CA, USA).

### 2.4. Procedure

The experiments were conducted individually in sound-proof rooms. Upon arrival, participants were told that the experiment concerned preference evaluations for people and food. The participants were instructed to fill out a set of questionnaires, including an assessment of eating habits and hunger levels, which took about 10 min. They were then seated 0.57 m from the monitor for the subliminal and supraliminal rating and forced-choice discrimination tasks.

For each of the subliminal and supraliminal rating tasks, 96 trials requiring preference evaluations (12 fast food, 12 Japanese food, 12 fast-food mosaic, and 12 Japanese food mosaic for both left and right visual fields) were performed in two blocks of 48 trials. Each block contained an equal number of trials for each stimulus type/food type/visual field condition. The order of conditions was randomized within each block. A short break was interposed between the blocks, and a longer break was interposed between the tasks. Participants initially completed five practice trials to become familiar with the procedure of each task.

In each trial of the subliminal rating task, a fixation point (a small cross) was first presented for 1000 ms in the center of the screen. Next, a prime stimulus was presented for 30 ms in either the left or right peripheral visual field (the inside edge was 5° from the center), followed by a mask stimulus presented for 170 ms in the same location. Then, the target ideograph was immediately presented in the center of the screen for 1000 ms. Finally, the rating display was presented until a response was recorded. Participants were instructed to gaze at the fixation point and rate their preferences regarding the target ideographs using a nine-point scale ranging from 1 (not at all) to 9 (very much) by pressing keys with the right index finger.

In each trial of the supraliminal rating task, a central fixation point (a small cross) was first presented for 1000 ms. Next, a target food/mosaic image was presented for 200 ms in either the left or right peripheral visual field (the inside edge was 5° from the center). After a blank screen had been presented for 1000 ms, the rating display was presented until a response was recorded. The participants were instructed to gaze at the fixation point and rate their preference for the target food/mosaic images in the same manner as in the subliminal rating task.

After the subliminal and supraliminal rating tasks had been completed, a forced-choice discrimination task was performed. A total of 48 trials were carried out using food images. In each trial, a food image was presented in the same manner as in the subliminal rating task. Then, two food images, one of which had been previously presented, were shown in the upper and lower visual fields. The two stimuli were in the same food subcategory. The participants were instructed to select the image that had been presented earlier. This task was based on the assumption that participants who had consciously perceived food images would be able to subsequently select those images.

Finally, interviews were conducted and the participants were asked whether they had consciously detected the primes during the subliminal rating task. Then, a debriefing was conducted. After explaining the research purpose, we requested the participants’ permission to analyze their data, which was granted in all cases.

### 2.5. Data Analysis

Data were analyzed using SPSS 16.0J software (SPSS Japan, Tokyo, Japan). The preference rating data for the subliminal and supraliminal rating tasks were analyzed separately. To simplify the analyses, differences in preference ratings between the food and mosaic conditions were calculated as the dependent measure. The preference difference scores were analyzed with two-way repeated-measures ANOVA, with participant culture (Polish, Japanese) as a between-subjects factor and food type (fast food, Japanese food) as a within-subjects factor. As our preliminary analyses showed that gender, age, BMI, and hunger level had no significant effects on the results, these factors were disregarded. The forced-choice discrimination data were analyzed using ANOVA with the same design, as well as one-sample *t*-tests contrasting with the chance level. The results were considered statistically significant at *p* < 0.05.

## 3. Results

### 3.1. Preference Ratings

Regarding the preference difference scores (food - mosaic) in the subliminal rating task (Figure 2 left), a two-way ANOVA with participant culture and food type as factors showed a significant interaction (*F* (1, 56) = 4.95, *p* < 0.05, *η*^2^*_p_* = 0.09). The main effects were not significant (*F* (1, 56) < 2.05, *p* > 0.10, *η*^2^*_p_* < 0.04). Follow-up simple effect analyses for the interaction revealed that the simple effect of food type, indicating higher preference for fast food than for Japanese food, was significant in Polish participants (*F* (1, 56) = 6.68, *p* < 0.05) but not in Japanese participants (*F* (1, 56) = 0.32, *p* > 0.10).

Regarding the preference difference scores in the supraliminal rating task (Figure 2, right), the two-way ANOVA showed only a significant main effect of participant culture (*F* (1, 56) = 4.67, *p* < 0.05, *η*^2^*_p_* = 0.08), indicating higher overall food preference in Japanese than in Polish participants. Other main effects and interactions were not significant (*F* (1, 56) < 0.56, *p* > 0.10, *η*^2^*_p_* < 0.02).

### 3.2. Forced Choice Discrimination

The mean ± standard error percentage correct responses of forced choice discrimination were 52.2 ± 1.9 and 52.3 ± 2.3% for Polish and Japanese participants, respectively. The two-way ANOVA using the same factors as above revealed no significant main effects or interactions (*F* (1, 56) < 0.60, *p* > 0.10, *η*^2^*_p_* < 0.02). One-sample *t*-tests revealed that the percentage of correct responses did not differ significantly from the chance level (*t* (116) = 1.65, *p* > 0.1). These results serve as an objective indication [31] that the primes had been subliminally presented in the subliminal rating task.

## 4. Discussion

Our results for the subliminal rating task revealed that Polish participants indicated higher preference ratings for fast food primes than for Japanese food primes, whereas Japanese participants indicated comparable preference ratings across these two conditions. This result corroborates the findings of previous cross-cultural studies reporting that participants’ cultural backgrounds moderate hedonic responses to food products, specifically, showing a lower preference for unfamiliar foreign food [9,10,11,12,13,14,15,16,17,18,19,20,21]. However, to date, no cross-cultural study has investigated unconscious hedonic evaluations of food. Understanding unconscious hedonic processing of food is important, because such processing has a major influence on daily eating behaviors [23]; moreover, it can be dissociated from conscious hedonic processing of food, as shown in this study and some previous studies [7,32]. This study provides the first evidence that participants’ cultural experiences moderate unconscious hedonic responses to food.

Unexpectedly, our results in the supraliminal rating task did not show an effect of food type on preference ratings in Polish participants. This result indicates that the conscious hedonic ratings can differ from rapid unconscious hedonic responses by adding some cognitive evaluations. It is known that Japanese food is generally healthier, and such information may have heightened conscious preferential evaluations of this food. Our results revealed higher preference ratings for food in Japanese participants than in Polish participants, which is consistent with previous findings [13,18] and may suggest a general tendency toward higher food preference in Asian participants compared with Western participants.

Our results have a practical implication. Understanding the influence of culture on hedonic processing of food is important for food product companies, as food trading exchanges have become globalized [33]. Our data suggest that culturally relevant food can strongly elicit unconscious hedonic responses in consumers. Because such cultural influence is supposedly due to familiarity gained through repeated exposure to the food [9,13,16,17,18,19,20,21], an increase in consumption opportunities may be required to induce hedonic responses to culturally new food materials. Furthermore, our data suggest that conscious preference ratings may not sufficiently demonstrate the moderating effects of culture on unconscious hedonic responses to food. Although food product companies typically rely on conscious self-reported ratings [34,35], implicit measures such as subliminal priming effects and assessment of emotional facial expressions [36] may be required to fully reveal rapid and unconscious hedonic responses.

Our results have theoretical implications for the broad literature of emotion processing. It has been proposed that unconscious emotional processing has effects on many aspects of daily life besides eating behaviors [37]. To date, only two previous studies have reported a moderating effect of participants’ culture on the emotional processing of unconscious stimuli [24,25]. However, because the stimuli used in those studies were images of the faces of people of other ethnicities, which can largely be processed based on innate programs [38], whether cultural learning experiences moderate unconscious emotional processing has remained uncertain. In contrast to facial processing, the existence of innate emotional representations of the food stimuli used in this study, such as hamburgers and sushi, is infeasible. Therefore, our results provide clear evidence that unconscious emotional processing can be activated by emotional representations acquired through cultural learning.

We speculate that a plausible neural substrate for the cultural moderation of unconscious hedonic responses to food may involve the amygdala. A recent neuroimaging study has demonstrated that the amygdala is activated in response to subliminally presented food images [39]. Several previous neuroimaging studies also revealed that participants’ cultural backgrounds moderated the activity of the amygdala during the processing of non-food stimuli, such as facial expressions [40,41]. In future research, it will be interesting to investigate neural activity associated with culture-dependent unconscious food processing so that we can understand the underlying neural mechanisms and develop implicit, objective measures for this psychological process.

Several limitations of the present study should be acknowledged. First, we only tested Polish and Japanese participants, and only used images of fast food and Japanese food. Hence, the generalizability of our results is unknown. Second, our sample size was small. Although we successfully detected the interaction between participant culture and food type, the analysis may have lacked the power to detect other important effects. For example, although we did not find any main or interaction effects of gender in our preliminary analyses, several previous studies reported that gender moderated hedonic food processing [42,43,44]. Finally, despite its potential relevance to the effects of cultural influence [9,13,16,17,18,19,20,21], we did not assess participants’ degree of familiarly with the food stimuli. Although none of our Polish participants reported regular consumption of Japanese food (as confirmed through interviews), any such participants who were more familiar with Japanese food may have been more likely to exhibit unconscious hedonic responses to images of that food. Relatedly, we did not assess participants’ degree of familiarly with the target ideographs, where familiarity may have affected the preference ratings, even though participants reported not being able to read the ideographs. Future investigations addressing these issues, for example by including individuals and food items from different cultures, and recruiting more participants and assessing their familiarity with the food of interest, should improve our understanding of the cultural moderation of unconscious hedonic responses to food.

## 5. Conclusions

Our results for the subliminal rating task revealed that Polish participants showed higher preference ratings for fast food primes than for Japanese food primes, whereas Japanese participants showed comparable preference ratings across these two conditions. In the supraliminal rating task, both Polish and Japanese participants reported comparable preferences for fast and Japanese food stimuli. These results suggest that cultural experiences moderate unconscious hedonic responses to food, which may not be detected using explicit ratings.

## Figures and Tables

**Figure 1 nutrients-11-02832-f001:**
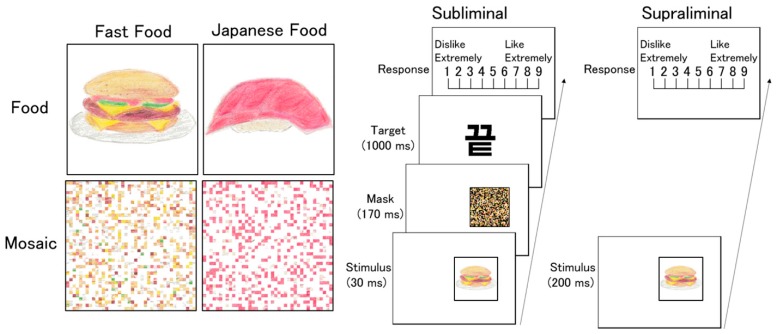
Illustrations of the fast food and Japanese food stimuli (**left**) and the trial sequence (**right**). The experiments employed photographic stimuli.

**Figure 2 nutrients-11-02832-f002:**
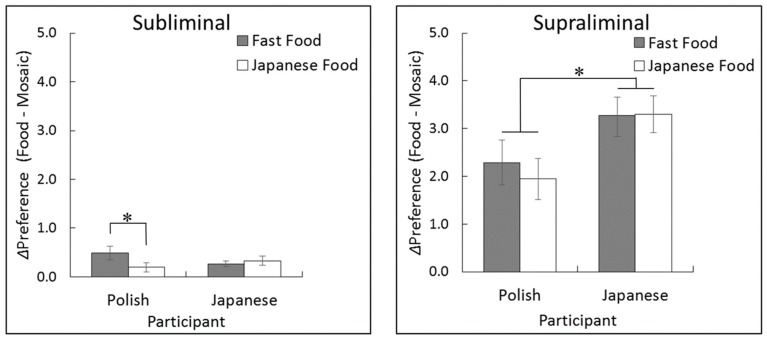
Mean (± standard error) preference difference scores (food versus mosaic) for the subliminal (**left**) and supraliminal (**right**) rating tasks. *, *p* < 0.05.

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
