# Peer review of "Cultural Moderation of Unconscious Hedonic Responses to Food"

_nutrients, 2019, doi:10.3390/nu11112832_

Round 1
Reviewer 1 Report
This manuscript investigates an important topic. When people see the sight of food, whether it is a “comfort food” or not, they feel the desire to consume it. The sight and smell of food elicits a conscious or unconscious desire for, and consumption of, that food. This particular work shows that participants’ cultural experiences modifies conscious hedonic ratings for food.
The paper is fun to read. However, I believe that there are certain drawbacks of the paper that should be mentioned as limitations. First of all, the number of participants in Poland and Japan is small. Second, we do not whether participants (especially the Japanese) were completely unfamiliar with the Korean characters that were presented to them as target stimuli. Third, we do not know whether any of the participants had travelled abroad and/or had some familiarity with different cultural representation of international cuisine. In other words, some of the participants (especially in Poland) could have been more “open-minded” than others when perceiving images of “foreign” food. This factor is not controlled in the experiment. Finally, the preferred term for the effect described by the authors is moderation, not modulation. Other than the points raised elsewhere in this review, no major weaknesses noted.
Author Response
We thank the reviewer for these helpful suggestions for improving our manuscript, which we have modified accordingly. Major changes to the manuscript are shown in red text. Additionally, we modified several descriptions in accordance with the Editor’s guidance, and language-related changes have been made by a professional English-language editing service (http://www.textcheck.com/certificate/ehSgKS); these changes are not highlighted unless they have altered the content.
Point 1
However, I believe that there are certain drawbacks of the paper that should be mentioned as limitations.
First of all, the number of participants in Poland and Japan is small.
Response
In accordance with the advice, we have noted the small sample size as a limitation of this study in the Discussion section (p. 16).
Point 2
Second, we do not whether participants (especially the Japanese) were completely unfamiliar with the Korean characters that were presented to them as target stimuli.
Response
As suggested, we have explained that our failure to assess the participants’ familiarity with Korean characters is a limitation of this study in the Discussion section (p. 16). Because we confirmed that the participants (both Japanese and Polish) were unable to read Korean characters before the experiment, we have added this information to the Materials and Methods section (p. 6).
Point 3
Third, we do not know whether any of the participants had travelled abroad and/or had some familiarity with different cultural representation of international cuisine. In other words, some of the participants (especially in Poland) could have been more “open-minded” than others when perceiving images of “foreign” food. This factor is not controlled in the experiment.
Response
We agree that the lack of assessment of familiarity with food materials is a limitation of this study. We now discuss this issue in the Discussion section (p. 16).
Point 4
Finally, the preferred term for the effect described by the authors is moderation, not modulation.
Response
As suggested, we changed “modulation” to “moderation” throughout the manuscript.
Reviewer 2 Report
I read this manuscript with great interest. Overall, this study was well designed and presented. I don't have major concerns.
Overall, this paper is of goog originality and quality. Some of my minor comments are listed below.
1) gender might be a factor that affects the outcome and this was not adequately addressed in the paper.
2) outcome variable measurements (scales) could be better explained.
3) normally, sample calculation is based on bi-variate analysis. When other factors are considered, sample size requirement tends to be larger. Thus, the small sample size seems to be a limitation.
Author Response
We thank the reviewer for these helpful comments to improve our manuscript. We have modified the manuscript accordingly. Major changes to the manuscript are shown in red text. Additionally, we modified several descriptions in accordance with the Editor’s guidance, and language-related changes have been made by a professional English-language editing service (http://www.textcheck.com/certificate/ehSgKS); these changes are not highlighted unless they have altered the content.
Point 1
1) gender might be a factor that affects the outcome and this was not adequately addressed in the paper.
Response
As suggested, we have noted that gender might be a relevant factor in the Discussion section (p. 16).
Point 2
2) outcome variable measurements (scales) could be better explained.
Response
In accordance with the advice, we have thoroughly described our rationale regarding the measurement of the outcome variable (i.e., preference ratings for the targets after positive versus neutral primes) in the Introduction section (p. 4).
Point 3
3) normally, sample calculation is based on bivariate analysis. When other factors are considered, sample size requirement tends to be larger. Thus, the small sample size seems to be a limitation.
Response
We agree that our sample may be too small to detect the influence of other factors, such as gender. We refer to this issue as a limitation of this study in the Discussion (p. 16).
Reviewer 3 Report
Strengths: Study appears to be well thought out. Appropriate controls in place. Well written. Clear and logical flow of ideas.
Suggestions:
-In the third paragraph of the discussion, the authors state that their data "suggests that domestic food products in one culture have less power to elicit unconscious hedonic responses in other cultures." From my interpretation of the data, the response of the Polish participants to the Japanese food was the same in both conditions; the only difference appeared to be a larger response to fast food in the subliminal condition. Consider revising this conclusion.
-Although the authors recognize the importance of familiarity in one of the closing paragraphs of the manuscript, in the remainder of the manuscript (eg., opening paragraph of the discussion) there appears to be an assumption that it was primarily cultural differences rather than familiarity that explained the differences in response between the two groups. I would like to see more commentary recognizing that factors other than the nationality of the participants (e.g., familiarity with the food) could explain the observed differences.
-Matching the y-axis scale on both figures in Fig 2 would add clarity and comparability.
Author Response
We thank the reviewer for these helpful suggestions for improving our manuscript, which we have modified accordingly. Major changes to the manuscript are shown in red. Additionally, language-related changes have been made by a professional English-language editing service (http://www.textcheck.com/certificate/C5R0ez), but these changes are not highlighted unless they have altered the content.
Point 1
In the third paragraph of the discussion, the authors state that their data "suggests that domestic food products in one culture have less power to elicit unconscious hedonic responses in other cultures." From my interpretation of the data, the response of the Polish participants to the Japanese food was the same in both conditions; the only difference appeared to be a larger response to fast food in the subliminal condition. Consider revising this conclusion.
Response
In accordance with this suggestion, we modified the description in the Discussion section to summarize our results (p. 14).
Point 2
Although the authors recognize the importance of familiarity in one of the closing paragraphs of the manuscript, in the remainder of the manuscript (eg., opening paragraph of the discussion) there appears to be an assumption that it was primarily cultural differences rather than familiarity that explained the differences in response between the two groups. I would like to see more commentary recognizing that factors other than the nationality of the participants (e.g., familiarity with the food) could explain the observed differences.
Response
As suggested, the Introduction (p. 3) and Discussion (p. 16) now underscore the importance of familiarity in relation to this issue.
Point 3
Matching the y-axis scale on both figures in Fig 2 would add clarity and comparability.
Response
As suggested, we created a new Figure that includes both the subliminal and the supraliminal results with matched y-axes. However, the figure showed that the means and SEs for the 2 tasks drastically differed, and the results for the subliminal tasks were difficult to present visually. This was because the supraliminal, explicit tasks (i.e., ratings for foods versus mosaics) produced much larger differences than did the subliminal, implicit tasks (i.e., ratings for ideographs primed by foods versus mosaics). For this reason, we created another Figure 2, in which both figures have comparable length of SE bars; we also included the above Figure in our Supplementary Figure 1.